# Cannabinoid Activity—Is There a Causal Connection to Spasmolysis in Clinical Studies?

**DOI:** 10.3390/biom11060826

**Published:** 2021-06-01

**Authors:** Daniel Joseph, Johannes Schulze

**Affiliations:** Institute of Occupational, Social and Environmental Medicine, Goethe-University Frankfurt/Main, Theodor-Stern-Kai 7, D-60590 Frankfurt/Main, Germany; da.joseph94@gmail.com

**Keywords:** tetrahydrocannabinol, dihydrocannabidiol, meta-analysis, central spasticity, evidence-based medicine

## Abstract

Cannabinoid drugs are registered for postoperative nausea and emesis, Tourette syndrome and tumor-related anorexia, but are also used for spasticity and pain relief, among other conditions. Clinical studies for spasmolysis have been equivocal and even conclusions from meta-analyses were not consistent. This may be due to uncertainty in diagnostic criteria as well as a lack of direct spasmolytic activity (direct causality). In this review we used the Hill criteria to investigate whether a temporal association is causal or spurious. Methods: A systematic literature search was performed to identify all clinical trials of cannabinoids for spasticity. Studies were evaluated for dose dependency and time association; all studies together were analyzed for reproducibility, coherence, analogy and mechanistic consistency. A Funnel plot was done for all studies to identify selection or publication bias. Results: Twenty-seven studies were included in this meta-analysis. The spasmolytic activity (effect strength) was weak, with a nonsignificant small effect in most studies and a large effect only in a few studies (“enriched” studies, low patient numbers). No dose dependency was seen and plotting effect size vs. daily dose resulted in a slope of 0.004. Most studies titrated the cannabinoid to the optimum dose, e.g., 20 mg/d THC. The effect decreased with longer treatment duration (3–4 months). The spasmolytic effect is consistent for different European countries but not always within a country, nor is the effect specific for an etiology (multiple sclerosis, spinal cord injury, others). For other criteria like plausibility, coherence or analogous effects, no data exist to support or refute them. In most studies, adverse effects were frequently reported indicating a therapeutic effect only at high doses with relevant side effects. Conclusions: Current data do not support a specific spasmolytic effect; a general decrease in CNS activity analogous to benzodiazepines appears more likely. Whether individual patients or specific subgroups benefit from cannabinoids is unclear. Further studies should compare cannabinoids with other, nonspecific spasmolytic drugs like benzodiazepines.

## 1. Introduction

Spasticity is a symptom of many diseases. In patients with multiple sclerosis its prevalence is ~85% [1], in patients with brain ischemia about 35% [2]. Due to pain and a lower range of motion, spasticity is debilitating [3,4] and eventually results in complete immobility and dependency [5]. Treatment of spasticity is empirical and complex. Since its pathogenesis is incompletely understood, its current treatment includes variable pharmacological approaches like GABA-ergic attenuation by benzodiazepines or baclofen, sympathic downregulation by tizanidine, or the peripherally-acting ryanodine receptor blocker dantrolen, in addition to pain relief and physiotherapy. Cannabis and its active compounds tetrahydrocannabinol (THC) and cannabidiol (CBD) have also been promoted; some studies suggesting a spasmolytic activity. In Germany, cannabinoids are licensed for the treatment of spasticity, postoperative nausea and vomiting (PONV), wasting in HIV and cancer and chronic pain syndromes [6], with postulated positive effects in Parkinson’s disease, schizophrenia, Tourette syndrome, epilepsia, chronic headache and chronic inflammatory bowel diseases [7]. Case reports and case series suggest the effectivity of cannabinoids also for other common diseases like depression or insomnia. These recommendations rest on systematic reviews [8] and indicate an association between cannabinoids and symptom relief. However, not all correlations are convincing, and they cannot prove a specific mode of action. Therefore, it is important to evaluate whether the observed spasmolytic cannabinoid effects are due to a general decrease in central muscle tonus or constitute a specific therapeutic approach differing from GABA activation or antisympathotonic effects.

Etiologically, spasticity has been classified as supraspinal, spinal and/or muscular [9,10]. Most theories consider lesions of the descending corticospinal [11] and reticulospinal motor tracts [12], with modulation of their activity by a variety of neurotransmitters and interneurons [13,14]. Selective isolated lesions in the pyramidal tract usually result in muscular hypotonia and hyporeflexia [14], whereas spasticity is more typical for lesions of descending pontine motoric neurons [12].

Spasticity is seen as a hallmark of central nervous system (CNS) diseases in contrast to atonic paralysis by diseases of the second motoneuron, efferent motoric nerves or muscle diseases [9]. It can be reliably diagnosed clinically but, like pain, not quantified, and has no generally accepted mechanistical explanation. The spasticity definition of Lance [15] is still accepted since no differences in pathophysiological etiologies could be established: this defines spasticity as an imbalance between muscular tonus and tendon reflexes, i.e., part of the upper motoneuron syndrome and, therefore, a CNS function. Other authors added muscular components like fatigue and decreased coordination [16,17], presence of cloni [4] and/or increased muscular sensitivity areas [18,19] as peripheral effects of spasticity although the focus on muscle contraction as the peripheral mode of action has been challenged [20,21].

Whereas there is consensus on the contribution of high muscular resistance to spasticity, the contribution of other neuronal or muscular components are variably weighed. A high basal muscle tonus [22,23] and decreased muscular elasticity [24,25] indicate the importance of peripheral factors, the increased excitability of stretch reflexes and the high basal activity of the first motoneuron (tonicity), indicate the importance of cerebral factors [12,26].

Spasticity is assessed by neurological examination. Measurements are semiquantitative and rely on patient rating scales (NRS = numeric rating scale) [27,28] or physician rating scales (modified) Ashworth scale AS or mAS [29,30]; both methods are widely in clinical use [31,32]. The AS has been validated [33] and has a higher reliability for the upper extremities [34], with low overall objectivity. The subjective spasticity NRS has been found equally reliable and valid [27]. Small changes can be overlooked especially in longitudinal measurements in both scales [9,35]. These properties explain differences between NRS and (m)AS in parallel measurements, with an overestimation of spasticity and therapeutic effects by NRS [27,36,37,38]. No clinical test can distinguish between etiologies blurring the pathophysiological causes of spasticity. They can be distinguished by apparative methods which are complex and laborious. These methods are less utilized, e.g., electromyography results in stronger signals upon passive stretching [39] with a lower signal threshold [40]. Its use is restricted to research and the correlation with the (m)AS is contested.

Prior meta-analyses of spasticity improvement by cannabis products, specifically (−)-trans-Δ^9^-tetrahydrocannabinol (THC) and (−)-trans-cannabidiol (CBD) binding to the cannabinoid receptors CB1 and CB2 [7,37], have inconsistently shown an association (yes: 6, no: 8, equivocal: 41). Since association is no proof of causality, but causality implies an association, we analyzed all relevant clinical studies and their synopsis whether they support causality in addition to association. For this, we applied Hill‘s criteria to answer the question whether cannabinoids improve spasticity in a distinct, specific mode. If the criteria are fulfilled it increases the likelihood of causality and may indicate a separate spasmolytic cannabis activity in addition to central activity attenuation.

## 2. Materials and Methods

A PubMED search was performed including all PubMED references before 1 January 2020, using the term “cannabinoids” AND “spasticity”. Non-English publications and letters, case reports, in vitro studies, reviews, studies using nonclinical endpoints or cost benefit studies were excluded based on the abstracts. From 57 articles no full text could be obtained. These articles, as well as reviews and open label studies were also excluded leaving 23 randomized, double blinded clinical trials (CT) or randomized clinical studies (RCT) with cannabinoid-based preparation effects on spasticity. From the citations in these studies, six other CT and RCT were identified. Since two of these studies reported identical patient groups, 27 RCTs or CTs were evaluated (see PRISMA diagram, Figure 1). Table 1 lists the study types included.

No distinction was made between studies using smoked cannabis, inhaling or orally taking cannabis medicinal extract (CME), tetrahydrocannabinol (THC) and/or dihydrocannabidiol (CBD). For further analysis, individual studies were analyzed whether they fulfilled those Hill criteria applicable to single studies. Some controlled trials used an “enriched study” design, i.e., only patients were included who had shown cannabis responses in a pretest, thus excluding all patients with severe adverse effects and/or no response to cannabis. These studies were included into the analysis but, since they were likely affected by a strong selection bias, their results are discussed separately. It was assumed that studies from one center or country likely included patients in more than one publication, and this selection bias also is discussed separately.

For studies using the NRS spasticity scale all data were presented on a 0–10 or 0–100 visual analog scale (VAS). For comparison data from other scales were transformed into a 0–10 scale if necessary.

To test for inhomogeneity, due to selective publishing, confounder effects, multiple inclusion of specific patients and using enriched studies, a Funnel plot analysis was performed [41]. If this is done plotting patient numbers versus effect size, a best fit line for extrapolating to ∞ patients approaching 1 indicates no therapeutic effect. If the effect size is plotted versus patient number deviations from the typical “funnel” appearance suggests an (unknown) selection bias and/or selective publication. Since a Funnel plot is not recommended for less than 10 studies, this plot was only performed for all studies together, not for subgroup analyses.

## 3. Results and Discussion

The best evidence for showing the spasmolytic effectivity of cannabinoids is a randomized, double-blinded clinical trial against placebo or a proven effective therapy. However, only a few studies were done as randomized studies using placebo as comparator. No study compared cannabinoids against standard spasmolytic therapy [42]. In most studies cannabinoids were used as add-on therapy (Table 1). Twenty-one studies included mainly or only multiple sclerosis (MS) patients. Riva et al. [43] and Weber et al. [44] treated patients with motoneuron-related spasticity (amyotrophic lateral sclerosis, ALS), and three studies [37,44,45] included spinal cord injury patients. Whereas MS is considered a demyelinizing disease indirectly affecting neurons, ALS as a motoneuron disease does not primarily affect myelin. Both diseases differ from spinal cord injury by their main location.

Spinal cord injury studies included low numbers of patients (Pooyania et al., 27 patients [45]; Hagenbach et al., 25 patients, [37]). The larger study by Berman et al. (106 patients [46]) reports decreased pain but no change in spasticity. A full text publication of this abstract has not yet been published. For motoneuron diseases like ALS only two studies with low patient numbers are published (Weber et al., 27 patients [44], Riva et al., 56 patients [43]). Riva found improved spasticity in the mAS, but not in other spasticity parameters (NRS spasm frequency, spasticity, 10 m walk test etc.). The effectivity in mAS was not confirmed by Weber et al. [44].

All other studies focused on spasticity by MS (Table 2). Among these, eight studies included >100 participants. With the exception of Novotna et al. [47] using an “enriched study“ design, no significant spasmolytic effect was seen (see Table 1). Some other studies with very few participants additionally measured fMRI or electrophysiological criteria [48,49,50]. Patient inclusion parameters varied according to these test methods. In one of the first studies of cannabinoids on muscle spasticity Fox et al. [51] found no improvement of dystonia, which also gives a positive (m)AS score. This study was not included. The lack of significant spasticity reduction is in line with the NICE review [8], whereas the reduction in spasticity is interpreted as “favors THC: CBD”, frequent and sometimes severe adverse effects favor placebo.

The lack of a consistent and significant effect does not exclude the possibility of a specific spasmolytic cannabis effect, possibly only for patient subgroups or spasticity from special etiologies. In order to investigate whether the published association may translate into a likely causality indicating a cannabis spasmolysis, we subjected all studies to causality criteria as described by Hill [52].

**Table 2 biomolecules-11-00826-t002:** Study characteristics for all included clinical trials of cannabis or cannabis extract studies on spasticity. MS: multiple sclerosis.

	Diagnoses	Patient Number; Location; Design	Drug, Doses, Duration, Application	Effect Size for AS or mAS
Killestein et al., 2002 [53]	MS	16 patients, pain center in NL	Oral 2.5–5 mg THC, 0–5 mg CBD, 4 wk	−5%, n.s.
Wade et al., 2003 [54]	diverse	21 patients, 1 center in UK, enriched	inhal THC, CBD, mix, 2 wk	−5%, n.s.
Zajicek et al., 2003 [55]	MS	657 patients; 33 pain clinics, UK, parallel group	Oral Cannador 2.5 mg THC: 1.25 mg CBD, 15 wk	−2%, n.s.
Vaney et al., 2004 [56]	MS	50 patients; rehabilitation center, CH, placebo controlled	Oral Cannabis extract, THC: CBD, 14 d	−20%, n.s.
Wade et al., 2004 [57]	MS	160 patients, 3 pain centers, UK, parallel groups	Inhal Sativex 2.7 mg THC: 2.5 mg CBD, 4 wk	−8%, n.s.
Zajicek et al., 2005 [58]	MS	355 patients; 33 pain clinics, UK, parallel groups	Oral cannabis extract 2.5 mg THC: 1.25 mg CBD, 52 wk	−20%, n.s.
Wissel et al., 2006 [59]	diverse	13 patients, 3 centers in Europe	Oral 1 mg THC, 4 wk	−35%, n.s.
Berman et al., 2007 [46]	SCI	106 patients, multicenter in UK, RO, parallel groups	Inhal Sativex, 7 wk	Not reported
Hagenbach et al., 2007 [37]	SCI	25 patients, 1 center in CH, parallel groups	Oral 2.5–10 mg THC, 6 wks	−47%, *p* < 0.001
Collin et al., 2007 [38]	MS	189 patients, 12 pain centers in UK and RO, parallel groups	Inhal. Sativex, 6 wk	−22%, n.s.
Conte et al., 2009 [49]	MS	17 patients, one clinic, I	Inhal. Sativex, 3 wk	0%, n.s.
Collin et al., 2010 [60]	MS	337 patients, 23 pain centers in UK and CZ, parallel groups	Inhal. Sativex, 15 wk	<7%, n.s.
Pooyania et al., 2010 [45]	SCI	27 patients, 1 center in CAN	Oral 0.5 mg THC, 4 wk	−12%, *p* = 0.003
Weber et al., 2010 [44]	Motoneuron diseases	27 patients, 1 center in CH	oral 2.5 mg THC, 2 wk	Not used
Novotna et al., 2011 [47]	MS	241 patients, 51 centers in Europe, parallel groups	Inhal. Sativex, 12 wk	Data not given, *p* = 0.094
Corey-Bloom et al., 2012 [61]	MS	30 patients, pain clinic UCSD, USA, placebo controlled	32 mg THC inhal., 3 d	−30%, *p* < 0.001
Notcutt et al.l., 2012 [4]	MS	36 patients, 5 centers in UK, withdrawal	Inhal. Sativex, 15 wk	Time to treatment failure, n.s.
Zajicek et al., 2012 [62]	Not specified	279 patients, 22 pain centers in UK, parallel groups	Oral cannabis extract, 2.5 mg THC, 0.8–1.8 mg CBD, 12 wk	Not used other methods
Langford et al., 2013 [63]	MS	42 patients, 11 clinic centers in F, CZ, enriched	Inhal. Sativex, 15 wk	Not used other methods
Tomassini et al., 2014 [50]	MS	18 patients, 1 clinic in I	Inhal. Sativex, 3 wk	+ 2%, n.s.
Vachova et al., 2014 [64]	MS	121 patients, 6 pain centers in CZ, parallel group	Inhal. Sativex, 48 wk	Not calculated, *p* = 0.212
Ball et al., 2015 [65]	MS	498 patients, 22 pain centers in UK, parallel group	Oral 3.5–7 mg THC, >1 year	0%, n.s.
Leocani et al., 2015 [48]	MS	34 patients, Italy, placebo controlled	Inhal. Sativex, 28 d	−25%; *p* = 0.041 for responders
Van Amerongen et al., 2018 [31]	MS	337 patients, 2 pain clinics NL, parallel groups	Oral 3.5–8 mg THC, 4 wk	0%, n.s.
Marcova et al., 2018 [66]	treatment resistant MS	106 patients, 16 clinic centers in A, CZ, enriched, parallel	Inhal. Sativex, 12 wk	7 muscles *p* < 0.05; 3 muscles *p* > 0.05
Riva et al., 2018 [43]	Motoneuron diseases	56 patients, 4 centers in Italy, parallel groups	Inhal. Sativex, 6 wk	−12%, *p* = 0.013

SCI = spinal cord injury; inhal. = inhalative; n.s.= not significant. Countries are indicated by their international country code (UK = United Kingdom; CH = Switzerland).

The Hill criteria [51] aid causality assessment when an association is found in epidemiological or clinical studies. As a foremost and important factor for causality, the strength of an association is estimated by the effect size and the specificity of the test used to measure spasticity. For spasticity, all tests include subjective judgements either by the patient (in NRS) or by the physician (in AS or mAS). Twenty-one studies used the AS or its modification, 16 used numerical rating scales; seven studies indicated an improvement in the AS, but only four had an effect size of ≥30% reduction (Figure 2). Since most studies found no or a small, nonsignificant effect, studies with a large spasmolysis warrant a closer look. Corey Bloom (30%, mAS [61]) screened 196 patients but enrolled only 37 patients. Markova (50%, mAS [66]) used an enriched study“ design, i.e., included only cannabinoid responders increasing the likelihood of a selection bias toward responders, and Zajicek et al. (50% [62]) used qualitative “response rates in categorial rating scales“, i.e., a derivative estimating the number of patients achieving a defined threshold. Wade et al. (35% [57]) studied 160 patients and found improvement in one numerical rating scale for spasticity, whereas 17 other tests, including mAS, could not confirm this finding. Hagenbach et al. (47%, mAS, NRS [37]) evaluated only 12 patients with spinal cord injury. Judging the data quality by the GRADE (Grading of Recommendations, Assessment, Development and Evaluations) criteria, as was done in the NICE analysis, all five studies had a moderate to very low quality to confirm spasmolysis [8]. 

Most studies found no (major) effect using the patient NRS scale or the physician AS scale, with a few exceptions. Most positive studies found a modest or low improvement in spasticity, with p-values between 0.01 and 0.05. A few other studies even found a tendency toward increased spasticity in some tests (Figure 1), or spasticity as an adverse effect [4,47,53,57]. In Figure 1, a best fit plot for all studies has a slope of −0.007; when excluding problematic studies (studies with effect sizes >30%) this slope further decreases to −0.0007, indicating the absence of a dose response relationship. Since all studies used multiple assessment methods, a correction for multiple testing [67] should have been included but is mentioned only rarely ([63], no significant effect, withdrawal study). 

It has been argued that cannabis effects manifest themselves only after an extended treatment time, as has been shown for chronic pain syndromes or mood disorders. Ball et al. [65] studied spasticity over three years, Zajicek et al. [58] over one year for volunteer patients. These studies used the longest follow up times, but neither study found a significant improvement. All other studies used 2–15 wk treatments, and time dependency was lacking, i.e., no changes in spasmolysis with treatment duration (data not shown). The inconsistency in the effect size among the studies, many negative studies, and methodological problems in all studies with a large effect (>30%) argues for a low effect size, if any, and thus does not favour a causal relationship. 

A biological gradient is present if a dose dependency can be shown. No single study used fixed CME, THC or CBD doses, or different dose groups. In most studies patients were asked to titrate their medication until a sufficient effect was achieved. This resulted in widely varying doses in most studies (usually ~20 mg/d THC on average), but no dose response range could be obtained from these data (Figure 2). 

Some studies reported an effect increase after prolonged treatment. Figure 3 plots the effect size against the treatment duration (two studies over one year [64] and three years [65] were omitted). The calculated slope for Figure 2 is −0.004, indicating no effect increase. 

Among the studies in Figure 3, those with the longest intervention used rather high daily THC/CBD doses but found only a small effect. Three studies with a similar duration of ~3 months [37,62,66] found a large effect size but with widely varying THC doses. Zajicek et al. [62] used 2.5 mg THC/1,25 mg CDB po, Markova et al. [66] 26.4 mg THC/23.75 mg CBD inhal, and Hagenbach et al. [37] used 31 mg THC orally over 6–10 weeks. Doses, application and THC combination with CBD were not consistent. 

Consistency of effects (also included in the GRADE guidelines) can only be tested with similar studies in different countries, and by different groups. Half of all studies were done in the UK, or in multiple countries including the UK, only two studies with low patient numbers were from outside of Europe (Cory-Bloom et al. [61] in the USA, Pooyania et al. [45] in Canada). Using different response criteria (mAS, NRS, spasm frequency) also decreases consistency among studies. In order to allow a better comparison, we tried to focus on spasticity as measured by the modified Ashford scale, or NRS. Even with this restriction, studies reported variable effect types like overall spasticity, spasticity in joint groups, or in the most affected limb, or calculated the number of participants fulfilling defined values (20% or 30% reduction). The last criterion, i.e., number of respondents, was used in those studies finding a high reduction. Zajicek et al. [62] reported a 50% increase in patients reporting a change in muscle stiffness, Markova et al. [66] reported a 20% improvement in spasticity, and Hagenbach et al. [37] used the mAS values for spasticity reduction, but only for seven patients. Including the fact that most studies did not achieve statistical significance, current data are not consistent and do not support specific or nonspecific THC or CBM effects on spasticity reduction.

Specificity, plausibility and coherence with current wisdom, are important factors for causality. Their lack, however, does not rule out causality. Cannabinoids bind to cannabinoid receptors with a plethora of pharmacological effects investigated for more than 20 years [68]. Specific spasmolysis by cannabinoids, if present, is no proof of CB receptor involvement in central spasticity (currently no physiological mechanism has been published for central spasticity), and the observation of catalepsy in CB1 receptor knockout mice argues against plausibility. Plausibility is no criterion, as long as no physiological explanation for spasticity exists. Coherence usually is assumed if cannabis has other effects relating to CB receptors and spasticity. It may also be assumed if the CB receptor location(s) concur with anatomical brain changes characteristic for central spasticity. Since spasticity is a physiological rather than anatomical concept, and cannabinoid receptor distribution cannot be determined clinically, coherence between the postulated spasmolytic affect and other criteria cannot be assessed. 

Clinical studies try to minimize placebo effects, usually by randomization. Randomization is problematic in cannabis studies, even in blinded studies up to 90% of the physicians could correctly identify treatment patients e.g., [50,53]. For participants it is not possible to blind inhalation studies since cannabis has a peculiar odor and smoking a placebo would be unethical. Thus, most studies have not specifically blinded the participants, some [4,47,54,63,66] used an “enriched“ design, only including patients with spasmolysis in a run-in period. Both factors may result in a higher rate of placebo responders. It also is possible that the recruiting procedure in specialized clinics preferentially recruits patients who have used cannabis before. To increase coherence within and between groups, cannabis effectivity should be compared to other spasmolytic agents. No comparative clinical study has been published to date.

Specificity for spasticity would also argue in favor of causality for cannabinoid effects. Spasticity is not well defined, can be measured only clinically and often is associated with other symptoms like pain, and both symptoms are cooperative and synergistic. Cannabis has been licensed for other indications unrelated to spasticity, like postoperative nausea and vomiting, Tourette‘s syndrome and wasting in cancer and AIDS patients and is under investigation for other diseases like Alzheimer’s disease, Parkinson’s disease and epilepsy [7]. The physiological role of the endocannabinoid system is poorly understood. For CB1 receptor knockout mice a higher mortality, decreased locomotor activity, hypoalgesia and catalepsy have been described, which is not in line with reduced spasticity as a specific THC effect [69]. Thus, specificity at the time being cannot be proven or refuted.

For epidemiological studies without intervention, criteria like temporality or experimental support are important, but are inherent in clinical studies. However, for cannabis, temporality should be seen critically, given its widespread recreational use. Only a negative urine test can rule out cannabis use outside of the study medication. A systematic urine screening at the beginning or during the intervention has been reported only in a few studies [31,48,49,53], casting some uncertainty on cannabis use outside of study medications for most studies.

Analogous effects by other cannabis receptor agonists would underscore the specific antispasticity by CBM or THC. Other cannabinoid-specific effects would also support causality. The lack of other physiologically based, specific, proven cannabinoid effects, besides lethargy, precludes using this criterion. Many studies reporting adverse effects list lethargy as the most common side effect, even if no spasmolysis can be seen. This supports the intake of the study medication but weakens a causal link between spasticity and cannabinoids.

The Hill criteria are intended to support or refute causality between two factors associated in clinical or epidemiological studies. Any association has to be validated before assuming causality. Forrest plots in meta-analyses are a tool to confirm and strengthen an association. They are no substitute for a separate causality analysis but help estimating the strength of an association. For cannabis, meta-analyses find, at best, a tendency for spasmolysis. Only temporality and partial experimental support favour causality but are inherent in clinical studies. Hill‘s criteria do not support a specific cannabis spasmolysis; thus, alternative explanations like nonspecific attenuation of CNS activity should be considered. 

Placebo effects or sampling confounders play a major role in study heterogeneity. A method to test for heterogeneity is a Funnel plot. The effect size, expressed as OR, is plotted against the standard error (SE) as an indicator for heterogeneity. For homogenous patient groups in different studies, all study results fall within a “funnel” created by the standard error. Major deviations (values outside the SE margins, lack of symmetry in the distribution) may be due to selective publishing (small cohort size) as well as unidentified confounders.

We plotted the OR of the (modified) Ashforth scale results against the standard error (Figure 4). Studies with a large (SE), i.e., small participant numbers, are published only with a positive effect, although some studies with negative results should be expected based on a statistical distribution. More important, for studies with larger patient numbers (SE below 0.2) six of 16 studies fall outside the expected funnel area. For the 16 studies included, only one study should fall outside this range, if any. One possible explanation is differing inclusion criteria resulting in noncomparable patient groups, as should be expected e.g., in “enriched studies”. 

## 4. Conclusions

Cannabis-based drugs have been licensed, often without evidence for therapeutic effects [6]. Although they are labelled for spasticity and used off label for multiple other ailments, meta-analyses (Table 3) failed to confirm spasmolytic effects. Significant effects were only achieved when enriched studies were included. In all studies a higher efficiency and higher significant results were seen with subjective scales like NRS rather that patient-independent scales like mAS. All positive studies reported high rates of adverse effects indicating pharmacological activity of the cannabinoids. 

Meta-analyses can support the strength of an association between cannabinoids and spasmolysis. Indicators for a causal connection e.g., by the criteria of Hill [51] would strengthen the assumption of therapeutic effectivity. Neither the Hill criteria based on study numbers (consistency), effect size nor the criteria for mechanistic likelihood were fulfilled, making a specific cannabinoid effect unlikely. If cannabinoids are spasmolytic, this likely is nonspecific. Cannabinoids should then be compared to other nonspecific spasmolytic agents like benzodiazepines or tizanidine, and their individual use should be based on a positive individual risk-benefit-ratio for cannabinoids and alternatives. 

## Figures and Tables

**Figure 1 biomolecules-11-00826-f001:**
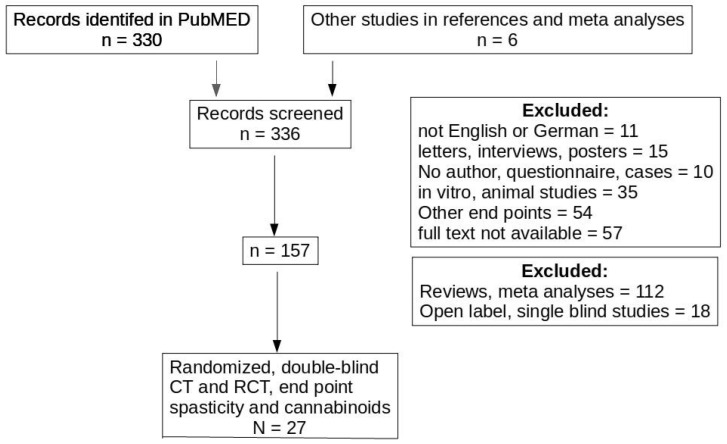
PRISMA diagram for literature identification. Three records fulfilled two exclusion criteria (language, other end points).

**Figure 2 biomolecules-11-00826-f002:**
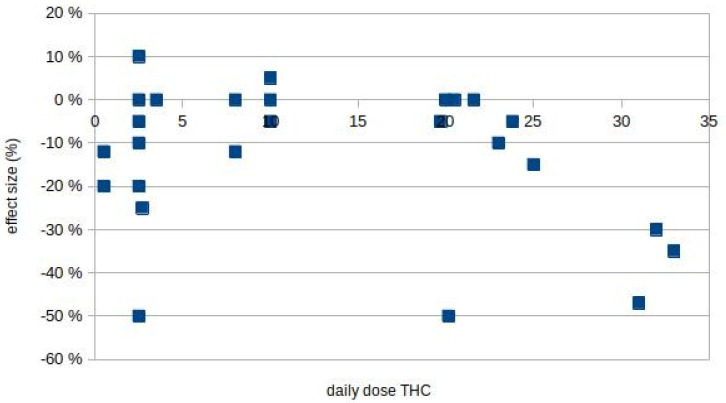
THC dose dependency of spasticity reduction, plotting spasticity reduction against the average daily THC dose (in mg). For discussion of studies with ≥30% improvement see text.

**Figure 3 biomolecules-11-00826-f003:**
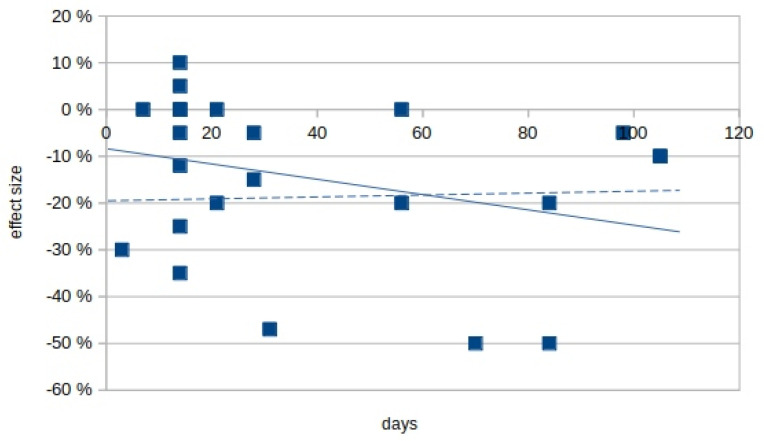
Spasmolytic effects by cumulative cannabinoid doses; the cumulative cannabinoid dose (daily dose, multiplied by days of application) is plotted against the effect size. For clarity, the studies of Ball et al. (65; 3 years) and Vachova et al. (64; 1 year) are omitted. — Best fit for all values; --- best fit for all values except [37,62,66].

**Figure 4 biomolecules-11-00826-f004:**
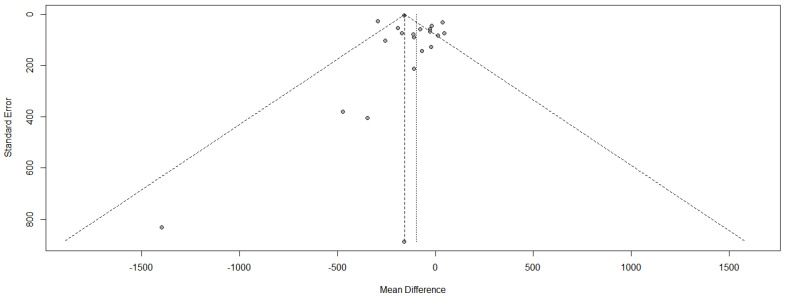
Funnel plot for heterogeneity among published studies.

**Table 1 biomolecules-11-00826-t001:** Included studies and type of spasticity specifically investigated. Three studies included two separate patient groups and, therefore, are listed twice.

Type	Study Number	Specifics
Placebo-controlled RCT	13	Multiple sclerosis spasticity
Placebo-controlled CT	9	Multiple sclerosis spasticity
Placebo-controlled RCT	2	Spasticity by mononeuron degradation
Placebo-controlled CT	1	Spasticity by mononeuron degradation
Placebo-controlled RCT	3	Spasticity due to spinal cord injury
Placebo-controlled CT	1	Spasticity due to spinal cord injury
Placebo-controlled CT	1	Spasticity due to stroke

**Table 3 biomolecules-11-00826-t003:** Meta-analyses for spasticity improvement by cannabinoids. * No information on significance due to low studies quality.

Author	Type of Cannabinoid	Number of Clinical Studies	Spasticity Criterion	Outcome
Whiting 2015	Nabiximols	5	mAS	n.s.
		3	NRS	improved
	Dronabinol	3	mAS	n.s.
	Cannabis extract	4	mAS	n.s.

NICE 2019	Nabiximols	7	mAS	improved *
		7	NRS	improved *
		4	NRS responder	improved *
	Dronabinol	3	mAS	n.s.
	Cannabis extract	3	mAS	n.s.
		1	NRS	n.s.
Da Rovare 2017	All cannabinoids	7	Not specified	n.s.
Torres-Moreno 2018	Cannabis extract	4	mAS	n.s.
	cannabis extract	2	subjective	improved
	nabiximols	7	mAS	n.s.
	nabiximols	8	subjective	improved
	dronabinol	2	mAS	n.s.
	dronabinol	2	subjective	n.s.

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
