# Peer review of "Cannabinoid Activity—Is There a Causal Connection to Spasmolysis in Clinical Studies?"

_biomolecules, 2021, doi:10.3390/biom11060826_

Round 1

Reviewer 1 Report

The manuscript is interesting and highlights how medicinal plants, in particular the cannabinoid active ingredients of Cannabis sativa, are helpful in therapy.

I suggest the authors to mention what are the active ingredients of Cannabis sativa, please look at this recent manuscript:

Gonçalves ECD, Baldasso GM, Bicca MA, Paes RS, Capasso R, Dutra RC. Terpenoids, Cannabimimetic Ligands, beyond the Cannabis Plant. Molecules. 2020 Mar 29; 25 (7): 1567.

In the introduction, the authors could highlight the importance of cannabinoids also towards other pathologies

Author Response

Thank you for your comments; we have included another sentence to highlight also the reports of effectivity for other common diseases, here depression and sleeping disorders. We have tried to restrict the manuscript to cannabis compounds known interact specifically with (brain) receptors, and also to known mechanisms of spasticity as a disease symptom. Therefore we did not include these compounds, but will certainly consider these when looking at other therapeutic cannabinoid effects.

Reviewer 2 Report

As cannabinoid drugs are used for spasticity and pain relief and clinical studies for spasmolysis have been equivocal, and even conclusions from meta-analyses were not consistent, in the present review the Authors have used the Hill criteria used to investigate whether a temporal association is causal or spurious.

Overall, I found this study timely, original, well conducted and scientifically sound. I have only some minor suggestions aimed to improve the high quality of the paper and these are outlined below:

1) I believe that in the introduction a brief note con cannabinoid receptors and activity would be useful to the reader with appropriate reference (see doi: 10.1177/039463200501800103).

2) The Authors conducted a Pubmed search. Please, specify the time frame.

3) Moreover, a PRISMA diagram for the literature searches would be vey welcome.

Author Response

Thank you for your valuable suggestions; we have included a short reference to CB receptors in the introduction, as well as a more specific reference at the appropriate position regarding to the Hill criteria specificity, plausibilita and coherence.

The time frame for the literature search is specified (before Jan 1st 2020).

We have included a PRISMA diagram as Figure 1.

Reviewer 3 Report

The subject is very interesting and worthy of being analyzed. Although the results that suggest not using any method or treatment seem to provide us with less data, it is also very important to know which therapies should not be taken into account. For this reason I consider that this article should be published.

However, there are several revisions that should be made:

  1. Changes on the presentation of an article for submission

            - In order to facilitate the possible changes that the reviewer detects or the submitter should correct, the lines of the article must be numbered.

            - Abbreviations require an explanation of their meaning the first time they appear in the text.

            - The references do not conform to the editorial standards of the journal.

            - There are several misprints throughout the manuscript, for example "nut" instead of "but" on page 9 paragraph 3. Please check the text to avoid these misprints.

  1. Changes in the expression or in the text.

            - There are confused concepts such as mixing spasticity of cerebral origin with that of spinal origin. Causes are also confused with effects, designating muscle spasticity as a type of cause when it is an effect.

            - One of the included studies of cannabinoids for spasticity had been performed for the treatment of dystonia (reference 52). It should be eliminated, since dystonia is not spasticity.

            - The presentation of the data obtained could be better organized to be more understandable.

Author Response

Thank you for your comments; numbering is activated in the second version.

We have checked for unexplained abbreviations (and found some); however, we have not explained “GABA receptor” or “HIV” since we consider these terms common knowledge in physicians or biologists.

We have checked for misprints (and hopefully found all).

In the context of CT or RCT studies spasticity is a clinical symptom, as we tried to outline in the introduction (lines 72 to 99 – we apologize for not having included the lines in the first version). We have included a further specification in lines 94 – 99 to emphasize the use of "spasticity" as a clinical symptom but not as an pathophysiological entity, and hope that this specification will be sufficient.

We agree that dystonia is etiologically different from spasticity. We included the study of Fox et al. since the authors also used the Ashworth scale; also, dystonia clinically is similar to spasticity and will give a pathological AS test. Restriction to spasticity as a neurophysiological phenomenon would exclude this study, complete coverage of all studies testing cannabinoids in patients with pathological AS spasticity values would favor inclusion. In favor of complete coverage we

- The presentation of the data obtained could be better organized to be more understandable.

would like to include the study in the Table; deleting this study would not change the results or the Figures.

We have organized the results according to Hills criteria with a brief overview of all CT studies of cannabinoids and clinical spasticity. We have tried initially to separate results and discussion but found this (traditional) format not as good as this format. However, we would welcome suggestions for a different format which enhances understanding.

All changes in the manuscript are highlighted in yellow.
